# UV-Vis quantification of the iron content in iteratively steam and HCl purified single-walled carbon nanotubes

**Markus Martincic, Gerard Tobías-Rossell** [ID] *

Institut de Ciència de Materiales de Barcelona (ICMAB-CSIC), Campus de la UAB, Bellaterra, Barcelona, Spain

* gerard.tobias@icmab.es

## Abstract

As-produced carbon nanotubes contain impurities which can dominate the properties of the material and are thus undesired. Herein we present a multi-step purification treatment that combines the use of steam and hydrochloric acid in an iterative manner. This allows the reduction of the iron content down to 0.2 wt. % in samples of single-walled carbon nanotubes (SWCNTs). Remarkably, Raman spectroscopy analysis reveals that this purification strategy does not introduce structural defects into the SWCNTs' backbone. To complete the study, we also report on a simplified approach for the quantitative assessment of iron using UV-Vis spectroscopy. The amount of metal in SWCNTs is assessed by dissolving in HCl the residue obtained after the complete combustion of the sample. This leads to the creation of hexaaquairon(III) chloride which allows the determination of the amount of iron, from the catalyst, by UV-Vis spectroscopy. The main advantage of the proposed strategy is that it does not require the use of additional complexing agents.

## Introduction

Nanotechnology offers new strategies to improve or even overcome some of the limitations of traditional materials [1–8]. Among the different types of nanomaterials, carbon nanotubes (CNTs) can be employed in a wide variety of applications that range from nanoelectronics, sensors, catalysis, composite materials and cancer therapy to name some [9–18]. The synthesis of CNTs results in samples that also contain both carbonaceous and non-carbonaceous impurities. These are typically amorphous carbon, fullerenes, graphitic onions, catalytic nanoparticles and catalytic particles. Furthermore, the catalytic particles could be coated by graphitic shells that protects them from direct dissolution. The presence of metal catalyst might not only induce toxicity [19,20] but, along with graphitic particles, can also affect and even dominate the electrochemical response of the sample [21–24]. High purity carbon nanotubes are indeed needed, for instance, for their application in the biomedical field to avoid undesired side effects [25–28]. The removal of impurities from the as-produced material is necessary for the application of carbon nanotubes in different fields, and various methods for this purpose have already been proposed [29–36]. The most widely employed strategies involve the use of strong

**Data Availability Statement:** All relevant data are within the manuscript.

**Funding:** Ministerio de Ciencia e Innovación, PID2020-113805, Dr. Gerard Tobías-Rossell Ministerio de Ciencia e Innovación, CEX2019-

000917-S European Commission, 290023, Dr. Gerard Tobías-Rossell.

**Competing interests:** The authors have declared that no competing interests exist

oxidizing acids, such as nitric acid, which can induce structural damage to the nanotube walls. Other strategies for the purification of carbon nanotubes include their treatment in oxidizing atmospheres such as air or oxygen, hydrogen peroxide, microwave heating, ultrasounds, electro-purification and magnetic separation to name some [37–40]. High temperature annealing has proven to be efficient in the removal of metal catalysts from multi-walled CNT samples [41], but it can pose a problem when dealing with single-walled CNTs because (SWCNTs), at high temperature, coalescence of SWCNTs has been observed forming multi-walled CNTs [42,43]. There is not a best strategy to purify CNTs, because it largely depends on the targeted application. For instance, the presence of functional groups introduced by nitric acid treatment, increases the dispersability and processability of the material and cross-links between the individual carbon nanotubes [44]. This can be an advantage or disadvantage depending on the final application. Acid treated CNTs are of interest for their further derivatization with functional groups, doping, and as anchoring points for the in-situ growth of inorganic nanoparticles [45–48]. Recently, Tanaka *et al.* reported on a damage-less dry-purification of carbon nanotubes using $FeCl_3$ vapor [49]. A low degree of structural defects is for instance desired for nanoelectronic devices and to prevent the undesired leakage of encapsulated materials when dealing with filled carbon nanotubes. We have placed our attention on the use of steam to efficiently eliminate impurities from carbon nanotubes without compromising their structural integrity and without introducing functional groups. Steam provides a gentle oxidizing atmosphere that effectively eliminates amorphous carbon and also graphitic particles, which might in turn contain metal catalysts in their interior. Following the steam purification, these metal particles become exposed (free from their protective graphitic coating) and can be readily dissolved in HCl [50]. Steam treatment has also been employed for the purification of highly aligned arrays of carbon nanotubes. A significant improvement in both quality, with respect to defect density, and in crystallinity, was observed resulting in an increased resistance to oxidation [51]. These super resilient carbon nanotubes were shown to withstand temperatures above 900°C under ambient conditions. The use of steam also results in the end opening of carbon nanotubes [52] thus allowing their posterior filling with a wide variety of payloads [53–55]. The carbon nanotube ends can then be sealed/closed thus allowing to remove the compounds that have not been encapsulated [56–59]. The confinement of materials inside the cavities of carbon nanotubes not only allows the formation of unprecedented structures but also expands their range of application [60–64]. Due to its gentle oxidizing properties, employing steam reduces the likelihood of introducing functional moieties or structural defects on the CNT structure [65].

Various techniques are employed to determine the amount of catalyst present in carbon nanotube samples, being inductively coupled plasma (ICP), thermogravimetric analysis (TGA), and SQUID the most frequently used. Agustina *et al.* reported on a colorimetric assay for quantifying iron in carbon nanotube samples [66]. In their study, the authors compared different acids for the complete digestion of the sample. Once digested, 1,10-phenanthroline was employed to form a red-orange complex with iron which could then be quantified. Herein, we propose a simplified approach that does not require the addition of a complexing agent to determine the amount of catalyst by ultraviolet-visible spectroscopy. The proposed strategy requires the use of a minimal amount of sample. For instance, the remaining residue after thermogravimetric analysis can be employed for its subsequent characterization by UV-Vis spectroscopy. UV-Vis is widely available in most laboratories and provides reliable data with a fast acquisition time. We also report for the first time a multi-step purification approach consisting on an iterative steam-HCl treatment that can effectively reduce the catalyst content in the samples.

## Materials and methods

Single-walled carbon nanotubes, *ca*. 200 mg, synthesized by chemical vapor deposition (CVD) (Elicarb, Thomas Swan Ltd.) were ground with an agate mortar and pestle. The ground SWCNTs were then introduced into a 4 cm silica tube, which was used as sample holder, and placed in the middle of a furnace. Argon was bubbled through hot water and the resulting argon/steam gas was introduced into the furnace and allowed to react with the carbon nanotubes at 900°C during for 4 h to eliminate amorphous carbon and graphitic particles. To dissolve the now exposed metal catalyst, the sample was treated with 200 mL of 6 M HCl (Panreac) at 110°C overnight. The sample was then washed several times with purified water until a neutral pH was achieved (by monitoring the filtrate) and subsequently dried at 80°C. Several purification steps were performed by repeating the argon/steam treatment followed by an HCl wash in an iterative process. To evaluate the effect of fractioning the steam treatment, a set of experiments was designed. In all of them the total amount of steam was kept at 6 hours. Namely, an iterative steam-HCl process was repeated six times (using 1 h steam), three times (using 3h steam), two times (using 3 h steam) and also a single time (6 h steam). The latter was included as a control. In all the cases an HCl wash was performed after each steam treatment to dissolve those metal particles that were exposed (after oxidation of the graphitic shells by steam).

Carbon nanotubes (CNTs), *ca*. 5 mg, were subjected to thermogravimetric analysis (TGA) using a STA 449 F1 Jupiter (NETZCH) under flowing air. TGA was performed by heating up to 900°C at 10°C min$^{-1}$ with a gas flow rate of 25 mL min$^{-1}$. The solid remaining after TGA contains oxidized iron, from the catalyst particles. This residue was recovered, dissolved in conc. HCl (50 mL), and employed for UV-Vis analysis to quantify the metal content (Cary 5.0, using 1 cm quartz cells).

Raman spectra was acquired between 100 and 2000 cm$^{-1}$ with a wavelength of 632 nm (20 mW He-Ne laser; Lab Ram HR 800 Jobin Yvon). For Raman measurements, samples were sonicated in isopropyl alcohol (Panreac). Few drops were then deposited on a glass substrate and the solvent was slowly evaporated at *ca*. 85°C.

SQUID measurements were conducted in a Quantum design MPMS XL-7T instrument at 10 K using an external DC field. A magnetic field range of -50,000 to +50,000 Oe was employed. For these measurements, SWCNTs (*ca*. 6 mg per sample) were placed inside gelatin capsules. Glass wool was employed to prevent any sample movement.

Scanning electron microscopy was performed in a QUANTA FEI 200 FEG-ESEM equipped with an energy-dispersive X-ray spectroscopy (EDX) detector. Sample for analysis was deposited on top of a carbon tape.

To quantitatively determine the amount of iron using UV-Vis analysis, a calibration curve was generated by measuring five different concentrations of iron(III) oxide, dissolved in conc. hydrochloric acid, within the range of 0.2 to 1 µmol/dm$^3$. To assess the amount of Fe in the SWCNTs after each treatment, the residue obtained after TGA was initially sonicated in few mL of conc. HCl. Subsequently, the volume was raised to 25 mL using volumetric flasks. The absorbance at 243 nm was fitted to the previously prepared calibration curve to determine the concentration of Fe in the different samples.

## Results and discussion

In this work we have employed a two-step purification strategy for SWCNTs that consists on an initial thermal annealing of CNTs under argon, in the presence of steam ($H_2O$), followed by an HCl treatment. Steam has been reported to open the ends of CNTs, and to remove the most reactive (defective) CNTs, amorphous carbon and graphitic particles. The graphitic shells

might in turn be coating catalytic NPs. After the steam treatment, the now exposed metal nanoparticles can be easily dissolved by HCl, the second step of the purification process. In fact, the acidic solution turns yellow due to the dissolution of the iron particles. The dissolved iron can subsequently be eliminated through filtration, and the SWCNTs can be recovered as a solid powder on top of the filter membrane.

Initially, we treated as-received SWCNTs for different periods of time with steam, followed by HCl. We will refer to this set of samples as "direct purification". As it can be seen in Fig 1A, the application of steam on SWCNTs for a given time, with a subsequent HCl wash, yields samples with varying levels of inorganic residue. This is attributed to the presence of iron in the samples, as previously determined in Elicarb® SWCNTs by X-ray photoelectron spectroscopy (XPS) analysis [50]. Initially, a decrease is observed up to three hours, followed by the subsequent apparent increase in the metal content. The minimum amount of iron (catalyst) was obtained after 3 h (1.1 wt% Fe) for this single-step steam treatment as determined by TGA (Fig 1A).

Iron in samples of SWCNTs can be quantitatively determined from the amount of residue present after TGA, using stoichiometric conversion. Iron undergoes oxidation to iron(III) oxide during the analysis in the presence of flowing air. As mentioned, after achieving the minimum metal content, an apparent rise of iron can be observed as the time of steam treatment increases. This phenomenon has been previously reported [50,52], and has been attributed to the fact that steam is more effective in oxidizing other carbonaceous species rather than the graphitic shells covering metal particles. TGA does not discern catalytic impurities from other inorganic compounds that can be present in the sample. Therefore, SQUID was next used to measure the amount of magnetic species, iron in the present study, in the samples. SQUID is considered an ultrasensitive technique that allows the quantitative determination of magnetic compounds in CNTs. The obtained data is presented in Fig 1B (solid black squares). In agreement with TGA, the obtained SQUID data also shows a decrease in the iron content after the purification treatment (steam + HCl) with respect to as-received SWCNTs. Noteworthy, the iron content remains practically constant, after 2 h of steam, within experimental error. Thus, the observed continuous increase of TGA residue, inorganic component of the sample, at 4 h, 5 h and 6 h might be due to the existence of additional inorganic compounds used in either the production of CNTs or their purification. Previous studies have identified the presence of alumina and silica in CNT samples produced through chemical vapor deposition [52,66].

In order to further reduce the iron content in SWCNT samples, the 6 h steam treatment was fractioned in two treatments of 3 h, with an HCl wash after each of them. Data shown in Fig 1B, indicated as "3-hour step" and denoted by solid red dots. A more significant reduction in catalyst content was observed with respect to the direct 6 h steam purification, even when the total treatment time was the same. This suggests a more efficient elimination of catalytic particles in the sample through the fragmentation of the purification process. Encouraged by this result, the steam treatment was further divided. The total exposure time of SWCNT samples to steam remained constant at 6 h, but the samples were collected either (i) every 1 h, from the steam purification system, (green solid triangles down-facing, Fig 1B) or (ii) every 2 h (blue solid triangles up-facing, Fig 1B). After each individual steam treatment, samples underwent an HCl wash. The amount of catalyst, in this case iron, was then quantified using SQUID. Notably, the lowest amount of Fe in the SWCNT samples was achieved by increasing the amount of iterations. This observation becomes most evident at the endpoints, in the samples that had received a total steam treatment of six hours (6-hour points in Fig 1B). The amount of catalyst could be reduced by approximately 40%, going from *ca.* 0.5 wt. % of iron with a single 6 h treatment to *ca.* 0.2 wt. % of iron by performing six independent steam treatments of 1 h. These results provide evidence of the enhanced effectiveness of the multi-step

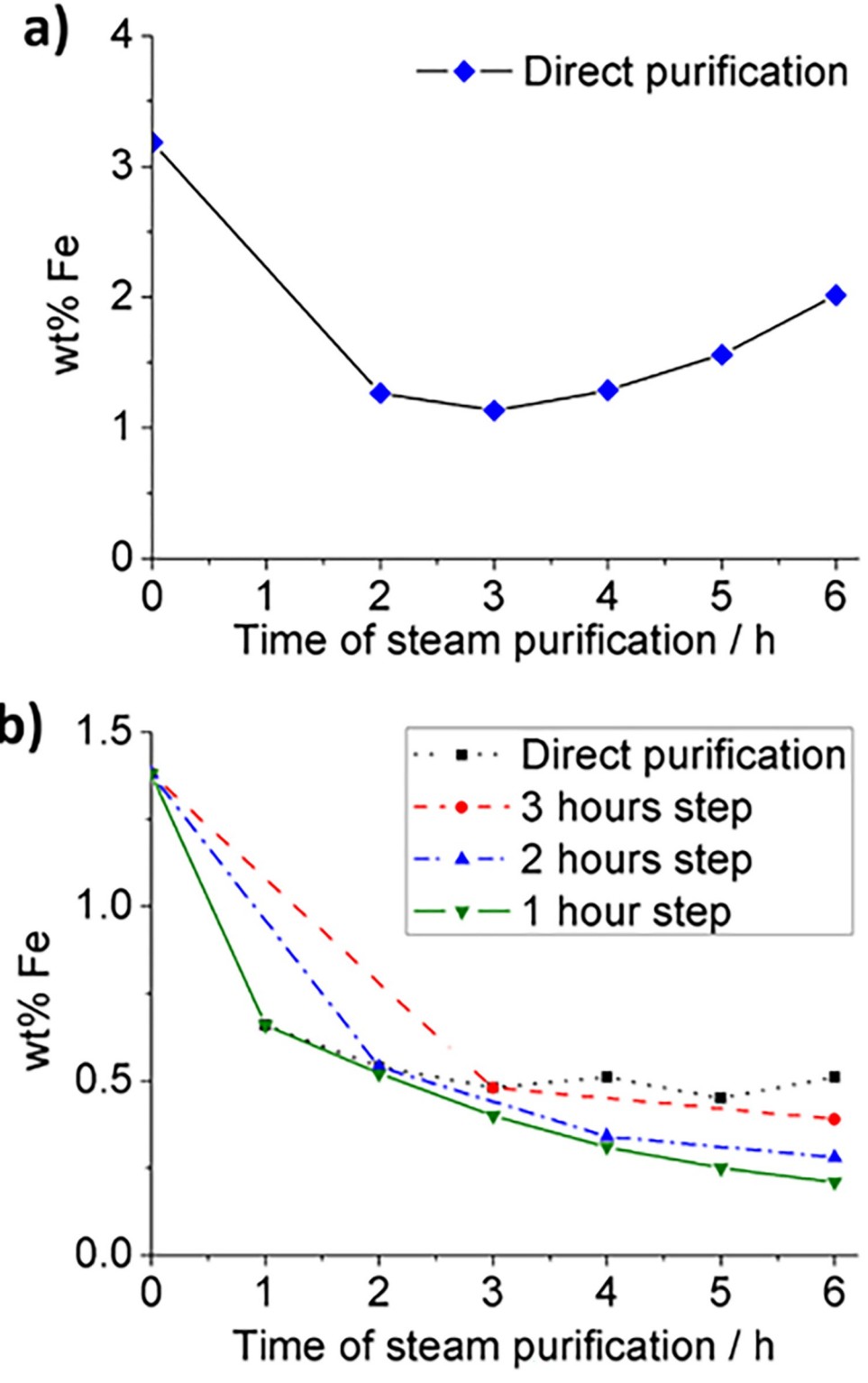

**Fig 1. Iron content in as-received and purified SWCNTs.** a) Amount of iron in samples of SWCNTs, which underwent a single steam treatment, "direct purification", as assessed by TGA; all values were recorded after treating the sample with HCl. Both as-received SWCNTs (0 h) and steam (2 h, 3 h, 4 h, 5 h, 6 h) and HCl purified SWCNTs are included. b) Amount of iron determined by SQUID for both as-received SWCNTs (0 h) and iteratively purified SWCNTs. In this case, the steam treatment was conducted either one time for a given duration ("direct purification") or in an iterative manner (1, 2, or 3 h steps), with a total steam treatment of 6 h. It is worth noting that two distinct batches of SWCNTs were used for the studies reported in (a) and (b).

treatment approach compared to the one-step "direct purification". The more iterations of steam + HCl are performed, using shorter steam treatment times, the lower becomes the amount of Fe in the samples. The reason behind this observation would need further investigation, but it could be due to a higher accessibility of steam to the whole SWCNT sample in the multi-step processing, because the material gets mixed during the acid treatment. Towards this end it would be interesting to perform the steam purification in a rotary tube furnace. Another possibility would be that a certain degree of reaction occurs between steam and SWCNT samples during heating and cooling of the furnace. This would result in slightly longer exposure times of the sample with steam when the purification is fractioned.

Next, we explored the use of UV-Vis spectroscopy for the quantitative assessment of the iron content in the prepared samples, as an alternative to SQUID and TGA, which are the most stablished techniques for the determination of catalyst in CNTs. Taking into account that hexaaquairon(III) chloride is readily formed when iron is dissolved in HCl, and that this iron complex can be directly quantified by UV-Vis spectroscopy, we investigated the use of UV-Vis for the quantitative determination of the amount of Fe present in SWCNT samples without the need of using additional complexing agents.

Initially UV-Vis calibration curves were prepared through the dissolution of known amounts of $Fe_2O_3$ in HCl. Fig 2 illustrates the recorded spectra alongside their respective calibration curves. Notably, three distinct peaks are evident. The peak at 243 nm, corresponding to the hexaaquairon(III), provides a more pronounced sensitivity:

$$y_{243} = 0.0836\ x + 0.0336;\ R^2 = 0.9993$$

than the peaks at 312 nm, and 360 nm:

$$y_{312} = 0.0549\ x + 0.0182;\ R^2 = 0.9992$$
$$y_{360} = 0.056\ x + 0.0186;\ R^2 = 0.9990$$

Therefore, this peak (243 nm) was employed to determine the amount of Fe in the samples.

To assess the sensitivity of the UV-Vis spectroscopic assay for the determination of the Fe content, all samples exposed to 6 h of steam were analyzed using this technique. This includes a single 6 h steam treatment (1 x 6 h) and also those samples that were subjected to multiple shorter steam treatments, namely 6 times 1 h (6 x 1 h), 3 times 2 h (3 x 2 h) and 2 times 3 h (2 x 3 h). Samples for UV-Vis analyses were prepared by treating the collected residue after TGA with HCl. The obtained solution was then analyzed using UV-Vis spectroscopy. Table 1 and Fig 3 provide a summary of the iron content as determined by the different techniques employed in the present study.

As observed, the iron content determined by UV-Vis closely aligns with the data obtained from SQUID (Table 1, Fig 3). Nevertheless, the amount of Fe determined by TGA is notably higher, as previously discussed. This difference could be attributed to the presence of other inorganic compounds in the samples, such as alumina or silica, that could have been introduced during the synthesis or purification of SWCNTs. The presence of such impurities contributes to the total amount of inorganic residue, which is detected by TGA.

Energy-dispersive X-ray spectroscopy (EDX) was next performed in the 6 x 1 h steam and HCl treated sample in a scanning electron microscope (SEM). Fig 4A shows a SEM image of purified SWCNTs (6 x 1 h) were several SWCNT bundles can be observed. EDX was performed in different areas of the sample, but due to the low amount of inorganic impurities it was difficult to assess the overall sample composition. An EDX analysis of the purified SWCNTs is shown in Fig 4B. In this specific area the presence of silica impurities can be detected, but this was not the case in other analyzed areas. To get a better overview of the sample composition, the 6 x 1 h treated sample was oxidized under oxygen to remove carbon and the resulting solid residue was analyzed by EDX (Fig 4C). As it can be seen in the analysis both Si and Fe appear to be present as impurities in the sample. The presence of oxygen is attributed

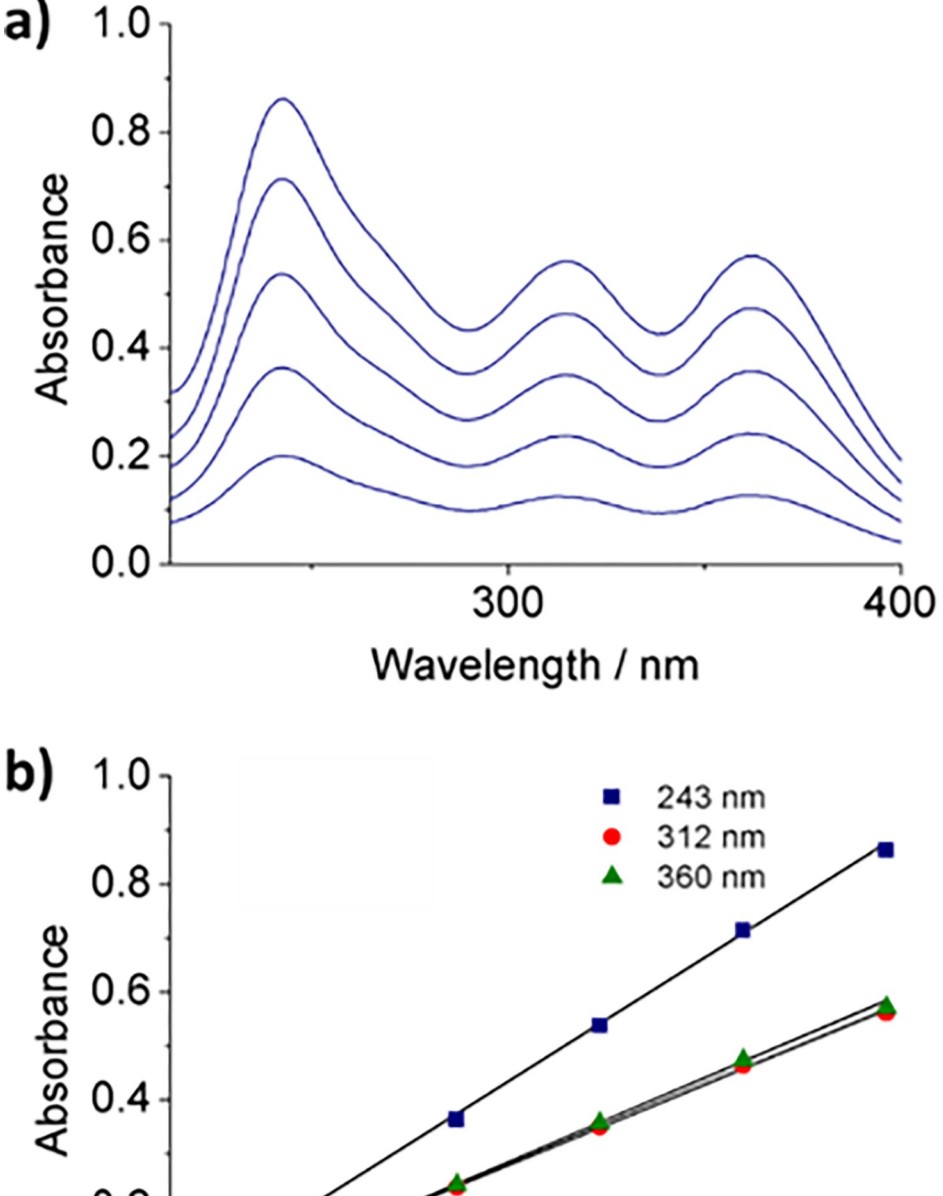

**Fig 2. UV-Vis calibration curves.** a) UV-Vis spectra obtained for the $Fe_2O_3$ standards and b) the resulting calibration curves using the recorded intensities of the three main peaks.

**Table 1. Amount of Fe determined by different techniques, and $I_D/I_G$ Raman ratio of SWCNT samples purified with steam for a total cumulative time of 6 h.** This includes a single 6 h steam treatment (1 x 6 h) and also samples that were subjected to multiple shorter treatments, namely 6 times for 1 h (6 x 1 h), 3 times for 2 h (3 x 2 h) and 2 times for 3 h (2 x 3 h). SQUID values correspond to the 6 h points shown in Fig 1B).

| Protocol (repetition x steam time) | TGA (Fe, wt. %) | SQUID (Fe, wt. %) | UV-Vis (Fe, wt. %) | $I_D/I_G$ |
|---|---|---|---|---|
| 1 x 6 h | 1,62 | 0,51 | 0,64 | 0,11 ± 0,04 |
| 2 x 3 h | 1,34 | 0,39 | 0,43 | 0,14 ± 0,04 |
| 3 x 2 h | 1,28 | 0,28 | 0,39 | 0,14 ± 0,01 |
| 6 x 1 h | 1,05 | 0,21 | 0,21 | 0,17 ± 0,07 |

to silica and to iron oxide, after the oxidation of the material. A carbon tape was employed as support. Silica and alumina have been previously observed as non-magnetic impurities in carbon nanotube samples, apart from graphitic particles and fullerenes [67], because these materials are widely employed during the growth and/or post-purification of the CNTs [52,66,68].

Even if silica was present during UV-Vis analysis, the quantitative assessment of iron using UV-Vis spectroscopy remains specifically associated with this element. The same principle applies to SQUID analysis, which exclusively detects magnetic materials and would be, therefore, insensitive to the presence of silica. Consequently, the UV-Vis methodology presented for the quantitative analysis of iron impurities in SWCNT samples, without the need of additional complexing agents, emerges as a straightforward, rapid, and cost-effective alternative to SQUID and TGA.

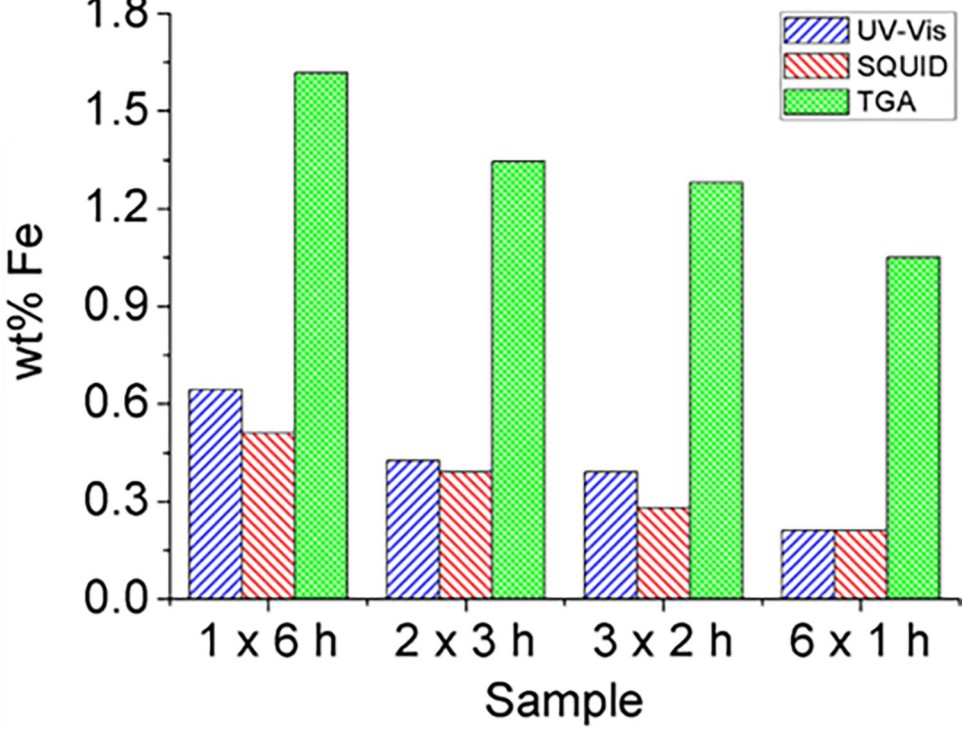

**Fig 3. Quantitative determination of the iron content in SWCNT samples.** Iron content determined by UV-Vis spectroscopy, SQUID, and TGA in purified SWCNT samples. These analyses were conducted following a cumulative 6-hour steam treatment, involving both direct purification (1 x 6 h) and multiple steam-HCl iterative purifications (2 x 3 h, 3 x 2 h, 6 x 1 h). An HCl wash was conducted after each steam treatment.

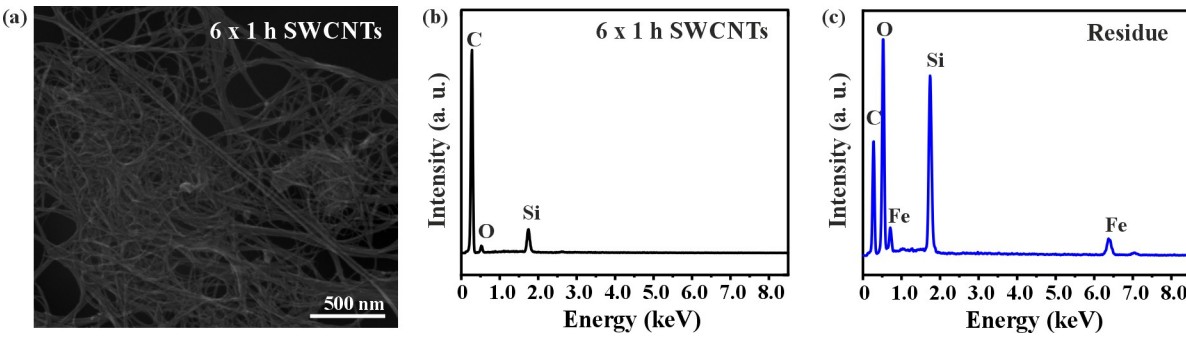

**Fig 4. Determination of morphology and residual impurities in 6 x 1 h steam and HCl purified SWCNTs.** a) SEM image of purified SWCNTs. b) EDX analysis of purified SWCNTs. c) EDX analysis of the solid residue collected after the oxidation of the purified SWCNTs.

To complete the study, Raman spectroscopy was employed to examine all the samples that underwent a total of six hours of steam treatment (1 x 6 h, 2 x 3 h, 3 x 2 h, 6 x 1 h). Table 1 provides the calculated D to G band intensity ratios, widely employed to evaluate the degree of structural defects in SWCNTs. The data in the table demonstrates that this ratio remains essentially unchanged, with variations falling within statistical error.

The combination of the above analyses confirms that a multi-step treatment is more efficient for the removal of iron in SWCNT samples, than a single-step treatment with steam for the same total amount of time. Furthermore, it is worth stressing than the multi-step treatment does not result in an increase in the number of structural defects. As mentioned, purified CNTs are of interest for a variety of applications including sensors or their potential use in the biomedical field. CNTs have been advocated as promising materials for drug delivery, biomedical imaging and targeted therapies. The use of nanotubes, and nanoparticles in general, in the biomedical field offer a unique platform to adjust essential properties such as solubility, diffusivity, pharmacokinetic profile, blood-circulation time and toxicity. Therefore, several nanoformulations are being explored in a wide variety of biomedical applications [69–75].

When it comes the quantitative determination of the amount of metal in SWCNTs, the present study complements the early work by Agustina *et al.* [66] on multi-walled CNTs thus validating UV-Vis spectroscopy as a fast an convenient method for the assessment of catalytic particles in both as-received and purified carbon nanotube samples. A simplified method is used herein that does not require the addition of complexing agents and buffers to adjust the pH of the solution. It also does not require the digestion of the sample in solutions of strong acids to oxidize the graphitic particles surrounding the catalysts. It is worth noting that the remaining residue after TGA, under an oxidizing atmosphere, is enough for a precise determination of the metal content. We believe that the proposed strategy is highly versatile and could be employed to evaluate metal content in other types of nanomaterials.

## Conclusions

In conclusion, we have shown that a multi-step purification approach, involving iterative steam-HCl treatments, is more efficient in reducing the amount of catalyst in SWCNT samples compared to a single steam and HCl purification. Analysis of the samples reveals that this approach does not introduce structural defects.

We have also proposed a simplified UV-Vis protocol for the quantitative assessment of Fe in samples of SWCNTs that consists on the dissolution, in HCl, of the solid remaining upon the complete combustion of the sample, thus eliminating the need of additional complexing agents.

## Acknowledgments

The authors would like to thank Thomas Swan Co. Ltd. for providing carbon nanotube material used for this study, and S. Sandoval and A. Cuesta (ICMAB-CSIC) for assisting in the SEM-EDX analyses of the samples.

## Author Contributions

**Conceptualization:** Markus Martincic, Gerard Tobías-Rossell.

**Funding acquisition:** Gerard Tobías-Rossell.

**Methodology:** Markus Martincic.

**Supervision:** Gerard Tobías-Rossell.

**Writing – original draft:** Markus Martincic.

**Writing – review & editing:** Gerard Tobías-Rossell.

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
