## [Decision Letter · Decision Letter 0]

12 Jan 2024

PONE-D-23-37335UV-Vis quantification of the metal content in iteratively purified single-walled carbon nanotubesPLOS ONE

Dear Dr. Tobías-Rossell,

Thank you for submitting your manuscript to PLOS ONE. After careful consideration, we feel that it has merit but does not fully meet PLOS ONE’s publication criteria as it currently stands. Therefore, we invite you to submit a revised version of the manuscript that addresses the points raised during the review process.

We look forward to receiving your revised manuscript.

Kind regards,

Amitava Mukherjee, ME, Ph.D.

Academic Editor

PLOS ONE

Journal Requirements:

Whilst you may use any professional scientific editing service of your choice, PLOS has partnered with both American Journal Experts (AJE) and Editage to provide discounted services to PLOS authors. Both organizations have experience helping authors meet PLOS guidelines and can provide language editing, translation, manuscript formatting, and figure formatting to ensure your manuscript meets our submission guidelines. To take advantage of our partnership with AJE, visit the AJE website (http://aje.com/go/plos) for a 15% discount off AJE services. To take advantage of our partnership with Editage, visit the Editage website (www.editage.com) and enter referral code PLOSEDIT for a 15% discount off Editage services. If the PLOS editorial team finds any language issues in text that either AJE or Editage has edited, the service provider will re-edit the text for free.

ERC Consolidator Grant NEST (725743)

Ministerio de Ciencia e Innovación, Spain (PID2020-113805 GB-I00)

Ministerio de Ciencia e Innovación, Spain (CEX2019-000917-S)

This work was supported by the ERC Consolidator Grant NEST (725743) and the Ministerio de Ciencia e Innovación, Spain (PID2020-113805 GB-I00 and CEX2019-000917-S). The authors would like to thank Thomas Swan Co. Ltd. for providing carbon nanotube material used for this study. 

ERC Consolidator Grant NEST (725743)

Ministerio de Ciencia e Innovación, Spain (PID2020-113805 GB-I00)

Ministerio de Ciencia e Innovación, Spain (CEX2019-000917-S)

7. Thank you for stating the following in your Competing Interests section:  

"No".

8. We note that your Data Availability Statement is currently as follows: All relevant data are within the manuscript and its Supporting Information files.

Reviewers' comments:

Reviewer's Responses to Questions

**Comments to the Author**

1. Is the manuscript technically sound, and do the data support the conclusions?

Reviewer #1: Yes

2. Has the statistical analysis been performed appropriately and rigorously? 

Reviewer #1: Yes

3. Have the authors made all data underlying the findings in their manuscript fully available?

Reviewer #1: Yes

4. Is the manuscript presented in an intelligible fashion and written in standard English?

Reviewer #1: Yes

5. Review Comments to the Author

Reviewer #1: Manuscript ID: PONE-D-23-37335

UV-Vis quantification of the metal content in iteratively purified single-walled carbon nanotubes

The manuscript entitled "UV-Vis quantification of the metal content in iteratively purified single-walled carbon nanotubes" was reviewed. This paper present a multi-step purification treatment that combines the alternated use of steam and hydrochloric acid. This manuscript can be accepted for publishing “PLOS ONE” but I have some major remarks before it can be publishable.

1. The abstract is unattractive. It is suggested that the importance of this research work be written in detail.

2. The title is informative and relevant, it could be more specific.

3. Authors have claimed “Carbon nanotubes (CNTs) can be employed in a wide variety of applications that range from nanoelectronics, sensors, catalysis, composite materials and cancer therapy to name some.”, I have read and evaluated the manuscript and in my opinion the submission does not yet sufficiently justify publication. The whole generalization for this paper should be given in the introduction. Discuss the shortcomings of previous work and the gaps and how this work intends to fill those gaps. Related references should be cited:

Materials Research Bulletin 48 (4) (2013) 1660-1667; Ultrasonics Sonochemistry 72 (2021), 105420; international journal of hydrogen energy 42 (39) (2017) 24846-24860; Polyhedron 28 (14) (2009) 3005-3009; Journal of Molecular Liquids 242 (2017) 447-455; Journal of alloys and compounds 617 (2014) 627-632; Diamond and Related Materials 79 (2017) 133-144

4. In the Abstract, the authors should emphasize what results the characterizations indicate.

5. The idea of the research seems to be interesting but the set goals are not achieved. What the main significance of paper in comparison is of relates published works? What is the innovation of this article? Please highlight it in the article.

6. Explain about the effective parameters which increase the yield of reactions?

7. Written is very week. In its current state, the level of English throughout the manuscript needs language polishing. Please check the manuscript and refine the language carefully.

8. Error bars in figures must be added.

9. It is suggested to evaluate the chemical composition of fabricated sample by other methods such as elemental mapping, to get clear evidence to readers for formation samples without impurities.

10. The Conclusions section is too long. It should be kept short and must be fully supported by the results reported.

I recommend this work for publication after above mentioned revisions.

6. PLOS authors have the option to publish the peer review history of their article (what does this mean?). If published, this will include your full peer review and any attached files.

Reviewer #1: No

---

## [Author Response · Author response to Decision Letter 0]

26 Feb 2024

Response to specific reviewer and editor comments have been included in the "Response to Reviewers" and "Cover letter" respectively.

---

## [Decision Letter · Decision Letter 1]

1 Mar 2024

PONE-D-23-37335R1UV-Vis quantification of the iron content in iteratively steam and HCl purified single-walled carbon nanotubesPLOS ONE

Dear Dr. Tobías-Rossell,

Thank you for submitting your manuscript to PLOS ONE. After careful consideration, we feel that it has merit but does not fully meet PLOS ONE’s publication criteria as it currently stands. Therefore, we invite you to submit a revised version of the manuscript that addresses the points raised during the review process.

We look forward to receiving your revised manuscript.

Kind regards,

Amitava Mukherjee, ME, Ph.D.

Academic Editor

PLOS ONE

Reviewers' comments:

Reviewer's Responses to Questions

**Comments to the Author**

1. If the authors have adequately addressed your comments raised in a previous round of review and you feel that this manuscript is now acceptable for publication, you may indicate that here to bypass the “Comments to the Author” section, enter your conflict of interest statement in the “Confidential to Editor” section, and submit your "Accept" recommendation.

Reviewer #1: (No Response)

2. Is the manuscript technically sound, and do the data support the conclusions?

Reviewer #1: (No Response)

3. Has the statistical analysis been performed appropriately and rigorously? 

Reviewer #1: (No Response)

4. Have the authors made all data underlying the findings in their manuscript fully available?

Reviewer #1: (No Response)

5. Is the manuscript presented in an intelligible fashion and written in standard English?

Reviewer #1: (No Response)

6. Review Comments to the Author

Reviewer #1: Manuscript ID: PONE-D-23-37335R1

I will consider publishing your paper entitled "UV-Vis quantification of the iron content in iteratively steam and HCl purified singlewalled carbon nanotubes"; authors present a multi-step purification treatment that combines the use of steam and hydrochloric acid in an iterative manner. After reading authors’ responses, I feel that my initial concerns have been partially addressed, but there are still some major concepts I cannot agree with.

Comments:

For Responses 1 and 2, I am grateful for additional information for the backgrounds.

For Response 3, it is suggested that this paper can provide more lists. The structure of the manuscript might need a major adjustment for a better understanding. To make more convincing, in the discussions, the authors should be considered some publication on the improvement of the explanation such as:

- Fuel, 351 (2023) 128885.

- Journal of Materials Research and Technology, 23 (2023) 3126-3136.

- Applied Surface Science, 255 (2009) 7610-7617.

- Chemical Engineering Journal, 173 (2011) 651-658.

- Separation and Purification Technology, 185 (2017) 140-148.

Journal of Molecular Liquids 242 (2017) 447-455; Journal of alloys and compounds 617 (2014) 627-632; Diamond and Related Materials 79 (2017) 133-144;

For Responses 4-8, I am grateful for additional information for the backgrounds.

For Response 9, in “Result and Discussions” section, the authors explained irrelevant and unnecessary subjects. For the characterizations, the results must be compared with other studies. You are requested to add some to amplify how this research work contributes to forwarding the field of study.

For Response 10, I am grateful for additional information for the backgrounds.

7. PLOS authors have the option to publish the peer review history of their article (what does this mean?). If published, this will include your full peer review and any attached files.

Reviewer #1: No

---

## [Decision Letter · Decision Letter 2]

24 Apr 2024

UV-Vis quantification of the iron content in iteratively steam and HCl purified single-walled carbon nanotubes

PONE-D-23-37335R2

Dear Dr. Tobías-Rossell,

We’re pleased to inform you that your manuscript has been judged scientifically suitable for publication and will be formally accepted for publication once it meets all outstanding technical requirements.

Kind regards,

Amitava Mukherjee, ME, Ph.D.

Academic Editor

PLOS ONE

Additional Editor Comments (optional):

Reviewers' comments:

Reviewer's Responses to Questions

**Comments to the Author**

1. If the authors have adequately addressed your comments raised in a previous round of review and you feel that this manuscript is now acceptable for publication, you may indicate that here to bypass the “Comments to the Author” section, enter your conflict of interest statement in the “Confidential to Editor” section, and submit your "Accept" recommendation.

Reviewer #1: All comments have been addressed

2. Is the manuscript technically sound, and do the data support the conclusions?

Reviewer #1: Yes

3. Has the statistical analysis been performed appropriately and rigorously? 

Reviewer #1: Yes

4. Have the authors made all data underlying the findings in their manuscript fully available?

Reviewer #1: Yes

5. Is the manuscript presented in an intelligible fashion and written in standard English?

Reviewer #1: Yes

6. Review Comments to the Author

Reviewer #1: Comments to Author.The authors have made great revision. It can be accepted now.

It can be accepted now.

7. PLOS authors have the option to publish the peer review history of their article (what does this mean?). If published, this will include your full peer review and any attached files.

Reviewer #1: **Yes: **NO NO NO

---

## [Editor Report · Acceptance letter]

29 Apr 2024

PONE-D-23-37335R2 

PLOS ONE

Dear Dr. Tobías-Rossell, 

I'm pleased to inform you that your manuscript has been deemed suitable for publication in PLOS ONE. Congratulations! Your manuscript is now being handed over to our production team.

Kind regards, 

on behalf of

Professor Dr. Amitava Mukherjee 

Academic Editor

PLOS ONE